# Adipose Tissue Plasticity in Response to Pathophysiological Cues: A Connecting Link between Obesity and Its Associated Comorbidities

**DOI:** 10.3390/ijms23105511

**Published:** 2022-05-14

**Authors:** Michelatonio De Fano, Desirèe Bartolini, Cristina Tortoioli, Cristiana Vermigli, Massimo Malara, Francesco Galli, Giuseppe Murdolo

**Affiliations:** 1Department of Internal Medicine, Endocrinology and Metabolism, Azienda Ospedaliera Santa Maria Misericordia, Ospedale di Perugia, Piazzale Gambuli, 06081 Perugia, Italy; michelantonio.defano@libero.it (M.D.F.); cristinatortoioli@libero.it (C.T.); cri.vermigli@gmail.com (C.V.); massimo.malara9@gmail.com (M.M.); 2Department of Pharmaceutical Sciences, Human Anatomy Laboratory, University of Perugia, 06132 Perugia, Italy; desiree.bartolini@unipg.it (D.B.); francesco.galli@unipg.it (F.G.)

**Keywords:** adipose tissue, adipose precursor cells, obesity, senescence, oxidative stress, type 2 diabetes, cardiovascular disease

## Abstract

Adipose tissue (AT) is a remarkably plastic and active organ with functional pleiotropism and high remodeling capacity. Although the expansion of fat mass, by definition, represents the hallmark of obesity, the dysregulation of the adipose organ emerges as the forefront of the link between adiposity and its associated metabolic and cardiovascular complications. The dysfunctional fat displays distinct biological signatures, which include enlarged fat cells, low-grade inflammation, impaired redox homeostasis, and cellular senescence. While these events are orchestrated in a cell-type, context-dependent and temporal manner, the failure of the adipose precursor cells to form new adipocytes appears to be the main instigator of the adipose dysregulation, which, ultimately, poses a deleterious milieu either by promoting ectopic lipid overspill in non-adipose targets (i.e., lipotoxicity) or by inducing an altered secretion of different adipose-derived hormones (i.e., adipokines and lipokines). This “adipocentric view” extends the previous “expandability hypothesis”, which implies a reduced plasticity of the adipose organ at the nexus between unhealthy fat expansion and the development of obesity-associated comorbidities. In this review, we will briefly summarize the potential mechanisms by which adaptive changes to variations of energy balance may impair adipose plasticity and promote fat organ dysfunction. We will also highlight the conundrum with the perturbation of the adipose microenvironment and the development of cardio-metabolic complications by focusing on adipose lipoxidation, inflammation and cellular senescence as a novel triad orchestrating the conspiracy to adipose dysfunction. Finally, we discuss the scientific rationale for proposing adipose organ plasticity as a target to curb/prevent adiposity-linked cardio-metabolic complications.

## 1. Introduction: Obesity Not Only a Matter of Fat Mass Accrual

The global epidemic of obesity represents one of the most important health hazards placing significant strain on healthcare systems worldwide [1]. The Global Burden of Disease study, a large systematic survey that analyzed data from 68.5 million persons at a population level in 195 countries over 25 years, found that an excess of body weight (as ascertained by high body-mass index, BMI) accounted for about four million deaths and 120 million disability-adjusted life-years worldwide in 2015 [1]. Yet, cardiovascular disease (CVD) and type 2 diabetes mellitus (T2D) emerged as the two leading causes of adiposity-related deaths. The cause-and-effect relationships between adiposity and the cardio-metabolic complications is better circumstantiated by the observation that these deleterious outcomes are regarded as the core defects of “metabolic syndrome”, a cluster of cardiovascular (CV) risk factors centered on abdominal obesity and insulin resistance (IR) [1,2,3,4].

Although obesity, by definition, represents a degree of excess weight associated with adipose tissue (AT) expansion, fat mass accrual *per se* does not appear to be the sole determinant of the adiposity-associated unfavorable cardio-metabolic outcomes. Different epidemiological trials show that some morbidly obese patients (and up to 25% of obese individuals) are metabolically healthy and, the other way around, about 18% of non-obese subjects demonstrate clinical features of IR syndrome [5,6,7]. However, the assertion that metabolically healthy obesity underlines a benign condition has recently been challenged [8,9]. A cohort study with prospectively collected data from “The Health Improvement Network” (THIN) database of approximately 3.5 million individuals, accruing 165,302 CV events during 5.4 years’ average follow-up, showed that subjects who were obese and “metabolically healthy” were still at an increased risk for coronary heart disease (49% increased risk), cerebrovascular disease (7% increased risk), and heart failure (96% increased risk), when compared with normal weight individuals with no metabolic risk factors [9]. Based on these data, the so called “healthy” obesity may likely be a misnomer as far as such a phenotype might underlie a lower “exposure time” to the adipose microenvironment, which, in turn, reflects a different plasticity of the adipose organ to expand in a healthy or unhealthy manner (see below).

It is thus counterintuitive that a better understanding of the inherent mechanisms regulating fat mass expansion (i.e., the adipose organ plasticity) are of paramount importance for the clinical readouts of systemic cardio-metabolic homeostasis [10].

## 2. The Adipose Tissue: A Remarkably Active and Plastic Organ

While underappreciated and misunderstood for a long time, AT is now regarded as a remarkably active and plastic organ with functional pleiotropism and high remodeling capacity [11,12,13,14]. The plasticity of the fat organ is unique as far as AT has the extraordinary capability of more than doubling in size via both hypertrophy (cell size enlargement) and/or hyperplasia (cell number increase), and then returning to baseline [15]. This physio-pathological cellular remodeling may either orchestrate nutritional adaptations in response to variations of energy balance to maintain biological stability (i.e., homeostasis) or leads to allostatic overload with unwanted disruption of both metabolic and cardiovascular control. In keeping with this unique plasticity, the adipose organ holds the key for improving our understanding of the obesity-associated cardio-metabolic dysfunctions.

In response to over-nutrition, AT expands by increasing triacylglycerol storage in adipocytes and undergoes dynamic, metabolic and cellular adaptive changes that protect remote organs from lipotoxicity. Rather than being a passive process, fat mass expansion is a sophisticated, energy-demanding program that is tightly regulated [16]. In obesity, AT remodeling may turn into a pathological process, which implies extensive changes in the ultrastructure of the organ involving the enlargement of existing adipocytes (i.e., hypertrophy), the formation of new fat cells from undifferentiated precursor cells (i.e., adipogenesis), extracellular matrix proteolysis, and the coordinated development of the tissue vascular network (i.e., angiogenesis). Nonetheless, within the expanded fat, complex cross-talks are engaged between different resident cell types through autocrine-paracrine pathways or cell contact, to which non-immune and progenitor cells contribute, accounting for metabolic and endocrine dysfunction of the unhealthy fat. Of note, despite such a remarkable plasticity to expand in a non-neoplastic manner, the capacity of AT to accommodate ongoing energy excess is finite, and the limit of fat mass expansion appears defined for any given individual [17]. Indeed, if insulin-resistance (IR) would be a direct consequence of increased fat mass, all the subjects would develop this metabolic complication at the same degree of adiposity. In contrast, on an individual level, there is no actual cut-point of adiposity as conventionally ascertained by the body mass index (BMI), clearly separating distinct insulin sensitive (healthy) from insulin resistant (unhealthy) obese sub-phenotypes [6]. In harmony with such an assumption, 29 and 80% of subjects classified as lean or overweight according to their BMI, respectively, were found to be obese, as estimated by their body fat percentage, and, more importantly, these individuals already exhibit a similar cardiometabolic risk as the obese patients [18]. Thus, although useful as an estimate of ‘‘adiposity’’ at the population-level, the *Quetelet index* should be regarded as an imperfect surrogate of adiposity at the individual level. In other words, BMI does not reflect the same degree of fatness according to sex, age, and ethnic origin of the subject and, hence, it does not accurately predict the health hazards of adiposity. At this purpose, a BMI-independent obesity staging system (i.e., the Edmonton Obesity Staging System, EOSS) was found to independently predict increased morbidity and mortality even after adjustment for contemporary methods of classifying adiposity, thus providing improved clinical utility in assessing obesity-related risk and prioritizing treatment [19]. This taxonomic upgrade partly challenges the previous concept of obesity that may be regarded as a disease, a condition, or a risk factor.

By far, individuals possess a genetically and/or environmentally determined limit for AT expansion, and the impairment of new adipose cell formation especially in the subcutaneous (sc) fat depots (i.e., abdominal and tight fat) has been suggested to be an instigator of the development of IR. This provocative paradigm (“expandability hypothesis”) partly explains the apparent paradox of an increased risk of IR in states of both expansion (i.e., adiposity) or reduction (i.e., lipodystrophy) of body fat [16,20,21]. It is thus tantalizing to speculate that adiposity and T2D represents two faces of the same coin, the insulin resistance syndrome, whose occurrence appears dependent upon the plasticity of the fat organ to expand in a healthy or unhealthy manner.

## 3. The Dynamics of AT Remodeling: Healthy and “Pathological” Expansion

As far as obesity poses extensive changes in the composition of the fat organ, an important distinction needs to be established between healthy and unhealthy or pathological fat mass expansion [12].

Basically, although proper fat mass expansion requires a highly coordinated response among many different resident cell types, including endothelial precursor cells, immune cells, and (pre)adipocytes, the AT mainly expands by a combination of adipocyte hypertrophy of preexisting cells and hyperplasia (i.e., recruitment and differentiation of new adipose precursor cells) [22]. These two features of fat expansion appear closely interwoven: the relationship between adipose cell size and obesity is indeed curvilinear, and when the increase in adipocyte size reaches a plateau, the generation of new adipocytes is triggered [22]. Elegant studies using ^14^C dating of adipocytes demonstrate that ~10% of the fat cells in human subcutaneous AT (SAT) are renewed every year, and a reduced regeneration capacity leads to inappropriate adipocyte hypertrophy (i.e., the so-called “hypertrophic obesity”) [23,24]. In line with this, Peter Arner and co-workers showed that total adipocyte number and adipose cell size in the SAT are negatively correlated and, in individuals with hypertrophic fat cells, the formation of new adipocytes is reduced [23,25]. Nonetheless, inappropriate hypertrophic expansion of SAT has also been reported in nonobese individuals with T2D as well as in nondiabetic subjects with a genetic predisposition for T2D (first-degree relatives; FDR) [25,26,27,28]. Since these features are seen long before T2D develops, adipocyte hypertrophy may well represent an early, obesity-independent, marker of IR and future risk of T2D driven by restricted adipogenesis and fat organ dysfunction (i.e., “adipocentric view”; vide infra). Hence, the inability to recruit new adipocytes “on demand” to store excess lipids in the SAT leads to hypertrophic (unhealthy) expansion of AT, while the ability to increase adipose cell numbers is protective, as postulated [29]. According to this rationale, healthy AT expansion consists of an enlargement of AT through the effective recruitment of adipose precursor cells to the adipogenic program, along with a proportional angiogenic response, appropriate remodeling of the extracellular matrix (ECM), and minimal inflammation. In contrast, a rapid growth of fat through adipocyte enlargement, a high degree of lymphocytes and macrophage infiltration, massive fibrosis, limited angiogenesis, and ensuing hypoxia characterize the pathological fat mass accrual [12]. In this scenario, the presence of healthy or pathological fat expansion is in line with the occurrence, in clinical practice, of the metabolically healthy or unhealthy obese sub-phenotypes, respectively [5,6,7] (Figure 1). Yet, the healthy or unhealthy signatures of fat may be interchangeable over time, also in response to pathophysiological or therapeutic cues, further highlighting the importance of the adipose organ’s plasticity for maintaining cardio-metabolic homeostasis.

## 4. Impaired Adipogenesis and Metabolic Dysfunctions: Role of Novel Lipids as Signaling Hormones (“Lipokines”)

### 4.1. Hypertrophic Obesity and Impaired Adipose Precursor Cells Differentiation: The “Revised” Expandability Hypothesis

Given the established link between fat organ plasticity and adiposity-linked cardio-metabolic injuries, it is tantalizing to hypothesize that insulin-resistance may be a pathological cue initiated in, and sustained by, a dysregulated fat (“adipocentric view”) [7,25,26,30,31,32,33,34,35,36,37,38,39,40]. Different, and not yet fully elucidated, biological signatures characterize a dysfunctional, unhealthy fat organ, such as: (1) the enlargement of existing fat cells (i.e., adipocyte hypertrophy); (2) the selective impairment of insulin signaling in adipocytes (i.e., the downregulation of the IRS-1 and the upregulation of the MAPK-dependent insulin signal); (3) the inflamed microenvironment with extensive macrophage and lymphocyte infiltration; (4) limited angiogenesis; and (5) hampered adipogenesis. The dynamics of these intricate events appears to be orchestrated in a cell-type, context-dependent and temporal manner. Mounting evidence strongly suggests that the failure to recruit and differentiate the mesenchymal adipose precursor cells (ASCs) may likely be the instigator for the development of unhealthy fat mass expansion [7,25,26,30,31,32,33,34,35,36,37,38,39,40]. In this context, the SAT plays a key role in the modulation of metabolic homeostasis both by acting as a metabolic sink for protecting remote organs against ectopic fat overspill (vide infra) and by preventing visceral fat accumulation in response to lipid overfeeding [6,41]. Rephrasing the original AT “expandability hypothesis” [42], which states that the capacity of an individual to expand fat mass to store lipids is a more important determinant of obesity-linked metabolic complications than the absolute amount of fat mass per se, it’s evident that the ability of SAT precursor cells to undergo adipogenesis represents a key mechanism regulating the capacity of AT expansion. Therefore, the characterization of the ASC in the context of healthy or unhealthy obesity has garnered a great deal of interest. Interestingly, Milan et al. recently proposed that ASCs reside in a discrete anatomical and functional unit called “adiponiche”, which works as a specialized and finely tuned in vivo *microenvironment* controlling cell fate and behavior [43,44]. Damage to one or more components (i.e., cellular and non-cellular elements) of the *adiponiche* results in a dysfunctional milieu that ultimately blunts the adipogenic potential of ASCs. Accordingly, the concept of stemness, rather than being an intrinsic property of a cell, is now held to be a feature strictly conditioned by the microenvironment [43,44].

### 4.2. Restricted Adipogenesis in the SAT Triggers Metabolic Dysfunctions

While the molecular aspects of adipogenesis are far beyond the purpose of this review [45,46], it is now clear that the individual ability to expand the AT is likely dependent on epigenetic, genetic and environmental factors [24]. Whatever the mechanism(s), restricted adipogenesis in hypertrophic obesity favors a diabetogenic milieu through different and partly synergistic mechanisms [47,48].

First, the inability to safely store metabolically active fatty acids derivatives in hypertrophic adipocytes leads to ectopic accumulation of lipids within non-adipose targets (i.e., skeletal muscle, liver, pancreatic islets, vascular endothelium, and myocardium) with detrimental consequences on tissue homeostasis (i.e., *lipotoxicity*) [49,50]. In harmony with this, a lipid-mediated endocrine network underlying the regulation of systemic metabolic homeostasis was recently postulated [51]. Experimental animal models and human studies [52,53] show that a single serum lipid (i.e., palmitoleic acid) may function as insulin sensitizing adipose-derived lipid hormone (i.e., lipokine). Further insights into this topic were recently gained by elegant studies published by Barbara B. Kahn’s and Ulf Smith’s groups [28]. By performing untargeted lipidomic analysis, a family of novel lipids, namely the branched fatty acid esters of hydroxy fatty acids (FAHFAs), was identified [54]. These novel lipids exert anti-inflammatory and anti-diabetogenic actions by enhancing insulin-mediated glucose uptake in adipocytes, glucose-stimulated glucagon-like peptide 1 (GLP-1) secretion from entero-endocrine cells and insulin secretion by pancreatic beta cells. Notably, AT from insulin sensitive mice overexpressing the glucose transporter GLUT4 had high levels of FAHFAs in contrast with the reduced abundance found in fat from insulin-resistant mice. The translational importance of these findings was also recapitulated in humans by the demonstration that: (1) FAHFAs are detectable both in fat and in serum; (2) isomers of one family of FAHFAs, palmitic acid esters of hydroxy fatty acids (PAHSAs), are reduced both in serum and AT of insulin-resistant compared to insulin-sensitive individuals; and, (3) serum PAHSA levels show a close positive correlation with the degree of insulin sensitivity. Interestingly, as far as heterogeneity in fatty acid composition between and within different fat depots has been reported [55], it would be of great interest to evaluate the occurrence and patho-physiological consequences of differences in FAHFA abundance between sc and visceral fat depot in humans. Our current understanding is thus that adipocyte acts as a glucose sensor [56] to modulate metabolic homeostasis. Although AT per se accounts for ~10% of insulin-mediated whole body glucose disposal, appropriate glucose flux into fat cells (mediated by GLUT4) is important to induce de novo lipogenesis (synthesis of fatty acids from glucose) and regulate the adipose secretome, the pattern of hormones which control metabolic and cardiovascular homeostasis. It is thus conceivable that the compartmentalization into adipocyte and the trafficking of specific fatty acids may either protect non-adipose organs from lipotoxicity or modulate insulin signaling at target tissues for preserving metabolic homeostasis.

Second, the enlargement of pre-existing adipocytes is responsible for AT microinflammation, instigating macrophage and lymphocyte infiltration. The inflamed AT secretes a pattern of prototypical adipose-derived hormones (i.e., adipokines) [39] that promote harmful cardiovascular and metabolic consequences associated with obesity [4,39,57]. Accordingly, increased macrophage infiltration and reduced circulating adiponectin levels characterize the dysfunctional and hypertrophic fat of the morbidly obese, but insulin resistant individuals (unhealthy obesity), when compared with similarly obese, insulin-sensitive, counterparts (healthy obesity) [7].

Third, the inflammatory microenvironment in inappropriately expanded fat organ negatively affects preadipocyte differentiation [58]. Locally released inflammatory molecules (i.e., TNF-a) can either sustain a macrophage-like phenotype in undifferentiated precursor cells [59] or also impair the ability of mature adipose cells to safely store triglycerides [60], thereby perpetuating a vicious cycle. On the other hand, experimental data on animal models challenged the concept that AT inflammation behaves as an anti-adipogenic cue [61]. In distinct mouse models with an AT-specific reduction in pro-inflammatory potentials, Wernstedt Asterholm and co-workers showed that the inability to sense and mount an appropriate local proinflammatory response at the level of the adipocyte reduces fat expansion leading to ectopic lipid deposition (i.e., hepatic steatosis) and metabolic dysfunction, implying that proinflammatory signaling in the adipocyte is actually required for healthy AT expansion and remodeling. Interestingly, the impairment of a proinflammatory response in the mesenteric (visceral) fat depot increased intestinal permeability, resulting in systemic inflammation and metabolic dysfunction even in the absence of a high-fat-diet [61]. Although the translational evidence for an inflammation-driven adipogenic action in human fat is still lacking, we can hypothesize a dual role of adipose inflammation in the regulation of fat organ plasticity: while a potent and transient inflammatory burst may be essential for proper AT remodeling and healthy expansion, the perpetuation of a chronic, low-grade, inflammatory response leads to profound changes in the *adiponiche* resulting in the development of a dysfunctional and unhealthy expanded fat organ.

Finally, increased oxidative stress and cellular senescence in the dysregulated fat further contribute to deteriorate the adipose microenvironment, hampering adipogenesis and tissue plasticity, as discussed below.

Taken together, these findings provide convincing evidence of the cross-talk between impaired adipose plasticity with restricted adipogenesis (i.e., hypertrophied adiposity) in a pathologically expanded fat and development of obesity-associated cardio-metabolic complications. Still, there is a clear need to further decode the molecular mechanisms that regulate AT expandability and to discriminate between normal physiological cues and those involved in metabolic syndrome. Unlocking these adipose-specific signals that impair recruitment and differentiation of adipose precursor cells appears attractive for biomarker discovery and targeted therapies [16].

## 5. Oxidative Stress and Cellular Senescence: Novel Actors in the Scene of Adipose Plasticity

### 5.1. Oxidative Stress and Lipid Peroxidation in Dysfunctional Fat: Oxysterols and 4-Hydroxynonenal as Novel “Lipokines”

Oxidative stress (OS), which results from the perturbation of a steady state condition where the free radical/reactive oxygen species (ROS) flux is balanced by antioxidant defenses [62,63], has importantly been involved in the pathogenesis of both atherosclerotic CV disease and IR/T2D [64,65,66]. The observation that the unhealthy AT is an important source of increased ROS production [65,66,67] harmonizes with the hypothesis that increased OS in accumulated fat is an early instigator of the obesity-associated cardio-metabolic complications [65,67,68]. The fine tuning of the redox balance in fat tissue may thus represent another important prerequisite for maintaining adipose organ plasticity through complex cross-talks involving different adipose cell-types. Basically, either mature adipocytes or adipose precursor cells are very sensitive to redox changes. Indeed, increased and uncontrolled production of ROS by enlarged dysfunctional adipocytes or infiltrating macrophages enhances OS in the fat microenvironment by establishing a vicious cycle that involves oxidative damage of locally stored lipids (i.e., lipoperoxidation) and impairment of precursor cell recruitment/differentiation.

Further insights into the complex relationships between adipose plasticity and impaired redox balance in the adipose microenvironment were recently provided by the characterization of lipid peroxidation [69].

*Lipid peroxidation* refers to the oxidative damage and degradation of lipids, a process initiated on polyunsaturated fatty acids (PUFAs) by ROS escaping the antioxidant system. By-products of lipid peroxidation may behave as toxic end products or also serve as signaling molecules, according to their exposure levels [70,71,72,73]. Such signaling effects primarily result from the adduct-forming capacity of these by-products with different macromolecules (i.e., proteins and nucleic acids), which ultimately causes structural damage and impaired biological activity [72,73,74]. AT is the main storage compartment of ready oxidizable lipids, and, thus, appears highly susceptible to OS and peroxidation. Redox lipidomic platforms have recently been developed to explore the occurrence, at the clinical level, of relevant indicators of lipid peroxidation and lipotoxicity [75,76]. Of note, vitamin E and their oxidation products were recently found to be of pathophysiological interest in obesity, IR, as well as in patients with non-alcoholic fatty liver disease (reviewed in [77]). These lipid metabolites, which can simultaneously be evaluated by means of targeted LC-MS/MS lipidomic protocols, include the lipoperoxyl radical-induced oxidation product of vitamin E and a-tocopheryl quinone [75,78]. The latter appears to be a very sensitive and specific indicator of lipid peroxidation, and a unique tool to explore the antioxidant effect of vitamin E on blood and cellular membrane PUFAs [79]. Among lipid peroxidation products, the non-enzymatic oxysterols and the hydroxyalchenals have also sparked considerable interest due to their mechanistic implications in the pathophysiology of obesity-linked cardio-metabolic complications [71,72,80,81,82,83,84,85,86,87].

*Oxysterols* are 27-carbon oxygenated products of cholesterol that may arise through enzymatic or non-enzymatic oxidation processes, or may be absorbed from the diet [62]. As ligands of sterol regulated transcription factors, oxysterols control gene expression in lipid metabolism, regulate immune and inflammatory responses, and modify cellular calcium signaling by acting at the transcriptional, translational, and post-translational level. A number of key proteins implicated in the control of metabolic homeostasis are recognized targets of oxysterol signaling [88,89,90,91,92]. Of note, by acting as a “sink” to safely store harmful cholesterol metabolites, AT plays a key role in regulating the trafficking of oxysterols and peroxidation by-products, an action that appears beneficial for preserving glucose homeostasis [93,94,95]. It is thus counterintuitive that the loss of these protective effects would reasonably result in local and systemic oxysterols “spillover” with deleterious effects. In line with such an assumption, serum concentrations of 7-ketocholesterol (7κ-C) and 7β-hydroxycholesterol, the most abundant OS-derived oxysterols that are carried on oxLDL [96], have been consistently associated with the development of T2D and coronary risks [84,97].

The importance of oxysterols in adipobiology stemmed from the characterization of oxysterols in human AT. In order to measure oxysterols as components of the human AT secretome, our group assessed by GC/MS analysis samples of interstitial fluid collected form the sc abdominal AT by the microdialysis technique [98]. Our data demonstrate the presence of 7κ-C and 7β OH-C in AT interstitial fluid at concentrations ≤1 µM. Moreover, isolated adipocytes from the SCAT of obese individuals with T2D showed increased abundance of these oxysterols as compared with non-diabetic controls. Finally, we found that 7κ-C and 7β OH-C at “physiological” concentrations (1 µM) impaired adipogenic differentiation, while the exposure of human adipose precursor cells at higher concentrations reduced cell viability/vitality. These observations highlight the concept that adipose precursor cells are the target of oxysterols, novel adipokines which are produced in vivo, accumulate in diabetic fat, and hamper adipogenic differentiation and fat tissue plasticity.

Besides cholesterol, PUFA may also undergo free radical-mediated oxidation generating a series of reactive carbonyls [73]. 4-hydroxynonenal (4-HNE) is one of the most studied highly reactive aldehydes formed during the non-enzymatic oxidation of lipid species containing n-6 polyunsaturated acyl groups. Different data in the literature support the theoretical likelihood that 4-HNE acts as a diabetogenic signal either by impairing insulin signaling/action in skeletal muscle or by blunting glucose-induced insulin secretion in b-cells and disrupting insulin’s effect through its direct adduction [82,83,99,100,101]. In keeping with this, the noxious effect of 4-HNE on AT homeostasis and development of T2D/IR have been clearly described in experimental models [86,102,103,104,105,106,107], as well as being recapitulated in humans. Jaganjac et al. reported impaired adipogenesis after treatment with 4-HNE in omental preadipocytes isolated from patients with T2D, providing a proof-of-concept that the synergistic actions of insulin and metformin may well reverse such a dysfunction [108]. Accordingly, by comparing paired samples of visceral and sc fat depots between obese metabolically healthy and obese “metabolically obese” premenopausal women, Jankovic et al. showed higher 4-HNE expression in the visceral depot of the latter vs. the former group, but roughly similar subcutaneous AT expression of 4-HNE, respectively [109]. These results point to 4-HNE as a potential marker of fat oxidation and dysregulation (i.e., diabetic adiposopathy), as well as a putative driver of impaired adipogenesis and systemic IR, as postulated [110].

Collectively, these data further circumstantiate the role of OS as a central player linking ROS and lipid peroxidation with macrophage infiltration and AT inflammation. Therefore, a rethinking of the entire issue of adipose lipoxidation with respect to macrophages should first be outlined, since these cells are the leading source of ROS, and byproducts of lipid peroxidation (i.e., 4-HNE) might themselves behave as potent chemoattractants for macrophages [111]. Nonetheless, the adipose progenitor cells have the potential to be trans-differentiated into macropahges [59], implying that a molecular cross-talk between (pre)adipocyte and macrophages mediated by lipid peroxidation may well provide a self-sustaining loop by which inflammation and OS conspire to impair AT plasticity and induce adipose dysregulation [39].

Another possible scenario can also be suggested whereby, while under physiological conditions, adipocytes provide a protective mechanism for removing and trapping harmful lipid peroxidation products; the increased formation and secretion of these novel lipokines induces profound changes in the *adiponiche* microenvironment, with unfavorable cardiometabolic outcomes. In this setting, it is however difficult to establish whether, chronologically, the cellular cross-talk orchestrated by lipid peroxidation first involves mature adipocytes, which behave as instigators of impaired precursor cells differentiation, or, alternatively, the increased abundance in fat of lipid peroxidation by-products formed in remote tissues/organs blunts adipogenesis with subsequent enlargement/dysfunction of existing mature fat cells.

### 5.2. Cellular Senescence: An Emerging Hallmark of the Dysfunctional Fat

Cellular senescence is defined as a cell fate characterized by a state of irreversible cell cycle arrest, combined with DNA damage and the induction of a senescence-associated secretory phenotype (SASP), which includes many inflammatory molecules, proteases, and miRNA’s [112]. Besides its association with ageing (i.e., “replicative senescence”), the senescent phenotype may also result from stress stimuli (i.e., “stress-induced premature senescence”), most often oxidative stress [113].

A bulk of evidence demonstrates that senescence is by far a central player in the development and progression of several chronic diseases including obesity, IR and T2D [114,115]. The “Geroscience Hypothesis” posits that manipulation of fundamental ageing mechanisms would delay (in parallel) the development of multiple chronic age-related diseases as a group, instead of one-at-a-time [116]. In this scenario, cellular senescence in fat emerges as a novel actor in the pathological adipose remodeling and altered cardiometabolic homeostasis. Indeed, white adipose tissue (WAT) appears highly susceptible to senescence either with ageing or independently of chronological age. According to the Mouse Ageing Cell Atlas, senescence seems to be initiated most early in fat than in other tissues/organs, and this leads to profound perturbations of organ plasticity and homeostasis [115,117]. Although the reasons underlying such a susceptibility of fat to undergo senescence remain unknown, different observations underscore the driving role of lipid peroxidation by-products (i.e., 4-HNE) in the induction of the senescent phenotype in different cell types [113,118,119]. As far as lipid peroxidation is a signature of the dysfunctional and “diabetic” fat, as previously outlined, it is possible to speculate that increased lipoxidation in fat might be a culprit of the higher susceptibility to senescence. This paradigm opens a novel scenario whereby inflammation, oxidative stress and senescence concur in the development of adipose dysfunction. Although the prevalent cell type(s) primarily contributing to AT senescence is still debated, the consequences of such a cell fate on different cell populations in fat may be clearly arguable.

First, senescent APCs/pre-adipocytes exhibit either a pro-inflammatory SASP or an impaired potential of differentiation, thus providing a “feed forward” mechanism between SASP and adipogenesis in promoting adipose organ dysfunction. Increased APC senescence may thus be a driver of the inappropriate hypertrophic adipose cell expansion and restricted sc adipogenesis either in patients with T2D and obesity or in non-obese FDR, as postulated [48]. Yet, since the process of senescence in APCs precedes the clinical occurrence of hyperglycemia, the possibility does exist that genetic and/or epigenetic factors controlling cell senescence susceptibility may contribute to the development of T2D and to the increased risk of patients with T2D for common ageing-associated chronic disorders, including cardiovascular (CVD) and Alzheimer’s disease [115].

Second, increased and premature senescence may also affect the endothelial cells, whose function is critical for the adipose microenvironment homeostasis either for allowing appropriate fat mass expansion in obesity (i.e., preventing AT hypoxia) or for supporting the lipid handling ability of fat [120]. In keeping with the latter, human AT microvascular endothelial cells have recently been shown to act as highly specialized cells by secreting, especially in response to certain fatty acids, peroxisome proliferator-activator receptor (PPAR-γ) ligands. The coordinated action of PPAR-γ activation (i.e., a master regulator for the lipid handling gene program) in endothelial and adipose cells enhances fatty acid uptake in endothelial cells and promotes lipid storage in mature adipocytes [120]. By impairing this “fine-tuned” regulation, senescent endothelial cells lose their ability to secrete endogenous lipid PPAR-γ ligands, which, in turn, could negatively affect pre-adipocyte differentiation and adipose cell function.

Third, besides the proliferative cells (i.e., APCs and endothelial cells), senescence also has a profound impact on mature (i.e., postmitotic) adipocytes [121]. Experimental data in animal models as well as recent human studies suggest that elevated DNA damage in adipocytes, which is a root cause of senescence in these cells, is active in the sc AT of severely obese subjects and precedes the development of obesity, AT inflammation, and glucose intolerance [114,122,123].

In this framework, it is therefore conceivable that AT senescence is another hallmark of the dysfunctional fat and an important player in the development of adiposity-associated cardiometabolic complications. Thus, targeting senescence selectively in AT may provide a novel potential therapeutic tool for preventing or/and treating different chronic diseases. Indeed, extensive experimental studies, as well as preliminary reports in humans, importantly suggest that treatment with “senolytics” (i.e., the combination of the kinase antagonist desatinib with the antioxidant quercitin), significantly decreases senescent cell burden in AT, improving adipose organ and metabolic dysfunction [124,125]. Yet, the discovery of “adipose-specific” senolityc agents, without safety concerns and with proven long-term effects on CV and metabolic homeostasis, is eagerly awaited.

## 6. Conclusions and Perspectives

The mechanisms underlying adipose organ plasticity are complex and still not fully clarified. The available literature provides robust scientific evidence indicating chronic inflammation, lipoxidation and senescence as key drivers of impaired adipose organ plasticity and obesity-linked cardiometabolic complications (Figure 2). This novel “triad” of the adipose dysfunction may well provide a holistic picture which reconciles the original expandability hypothesis, mainly centered on adipocyte “expansion limit”, with the oxidative, inflammatory and senescence theories, focused on changes in the *adiponiche* microenvironment. Although the exact chronobiology by which each component of the triad orchestrates the dynamics of adipose dysfunction remains to be defined, we outlined an intriguing scenario whereby the impairment of APC differentiation emerges as a connecting link between unhealthy fat mass expansion and the development of obesity-associated cardiometabolic outcomes.

Notably, unlocking the inherent adipose-specific mechanisms underlying all the elements of the triad is attractive for targeted therapeutics. Attempts to crudely use antioxidant supplementation, anti-inflammatory drugs or senolytics as therapeutic means for reverting adipose dysregulation and improving cardiometabolic complications in obesity and T2D is a dynamic filed. A number of human studies have found that the anti-oxidant α-lipoic acid, glutathione, vitamin E, vitamin C, and flavanols may ameliorate insulin-sensitivity in patients with IR, T2D and/or cardiovascular disease [116]. Moreover, antioxidants, including melatonin, the main product secreted by the pineal gland, could play a protective role in the development of diabetes and its cardiovascular complications partly by preserving stemness and reducing cell senescence of APCs through its effects on ROS generation [126,127]. In addition, the action of “senolytics” on both the inflammatory and senescent cell burden in AT [124,125] further highlights the possibility that “adipose-specific” agents may possibly reverse all the elements of the adipose triad hopefully without safety concerns and with proven long-term effects.

Wish as we might, the possibility that such a strategy can be successful for preventing, reversing or halting the progression of adiposity-linked cardiometabolic complications is promising. We need to live and learn more.

## Figures and Tables

**Figure 1 ijms-23-05511-f001:**
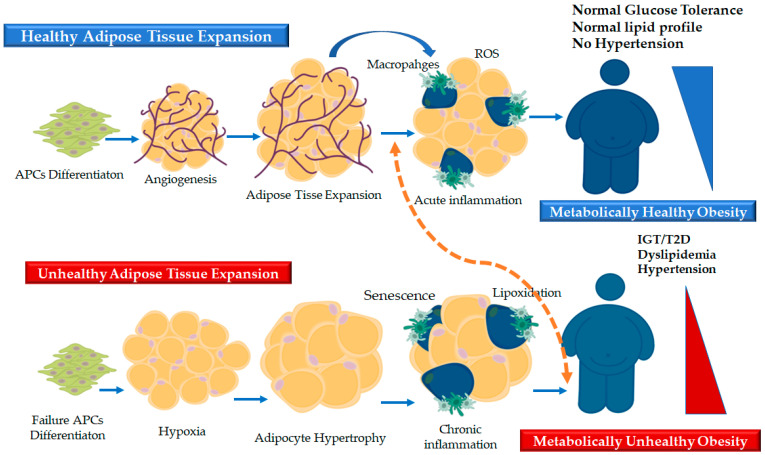
Adipose organ plasticity. “Healthy” or “unhealthy” signatures of fat expansion and occurrence of “metabolically healthy” or ‘‘unhealthy’’ obese sub-phenotypes. The “healthy” or “unhealthy” signatures of fat may be interchangeable over time (dashed arrow).

**Figure 2 ijms-23-05511-f002:**
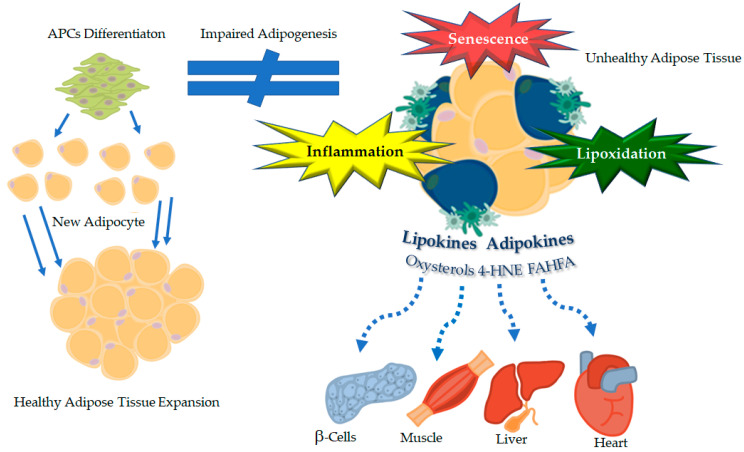
The novel “triad” of dysfunctional fat, inflammation, lipoxidation and senescence, as a key driver of obesity-linked cardiometabolic complications. APCs, adipose precursor cells; 4-HNE, 4-hydroxynonenal; FAHFA, fatty acid esters of hydroxy fatty acids.

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
