# Peer review of "Adipose Tissue Plasticity in Response to Pathophysiological Cues: A Connecting Link between Obesity and Its Associated Comorbidities"

_ijms, 2022, doi:10.3390/ijms23105511_

Round 1

Reviewer 1 Report

The current review by De Fano et al is well written, interesting and well worth publishing. However some key references are missing and some concepts could be clarified and expanded a bit before publication:

  1. Given the focus of the review and the many mentions of the Adipose tissue Expandability hypothesis, the original reference for it (Virtue & Vidal-Puig: It's Not How Fat You Are, It's What You Do with It That Counts, Plos Biology, 2008) should be references to – I believe it is missing from the current reference list.

  1. The Expandability hypothesis is, both in the abstract and on for example on row 121, called novel. However, it has existed for over 10 years in the literature (see reference above) making it not that novel anymore. The current paper would therefore benefit from the authors going through it and reserving the word novel only for concepts that have emerged in the last 1-2 years. One can instead talk about a shift in acceptance or attitude by the field when a theory has been there but not readily accepted in the beginning, if that is what the authors meant?

  1. According to my interpretation, the original Adipose tissue Expandability hypothesis was, as the authors call it, “adipocentric” and focused on the capacity of adipocytes to store lipids, and less of the role of adipogenesis. Therefore, the concept of defective adipogenesis limiting the adipose tissue’s capacity for expansion is a separate, emerging hypothesis about how obesity and insulin resistance are linked, that has come from the original expandability hypothesis. It seems to me that this concept is different from the original adipose tissue expandability hypothesis, and would therefore require a different name, either already proposed in other papers or laid forward in the present review. The current paper mixes the use of this name for both the original (2008) hypothesis and the renewed, refined one focusing on adipogenesis, making it unclear.

  1. Figure 1 is copied from another publication, as clearly stated in the figure legend. I am however sceptic that this can be done without asking permission, and think a good publication should preferentially contain original artwork, not copied ones.

  1. Row 184-184: doesn’t the SAT do both things (protect against ectopic fat and against spillover to visceral fat)? I suggest changing EITHER on row 184 to BOTH, and OR on row 186 to AND.

  1. The authors mention that inflammation perturbates adipogenesis – however I think that concept is very much dependent on timing and context, and the authors should include and also discuss the paper from the Sherer lab (Wernstedt Asterholm I, Tao C, Morley TS, Wang QA, Delgado-Lopez F, Wang ZV, et al. Adipocyte inflammation is essential for healthy adipose tissue expansion and remodeling. Cell Metab. 2014;20(1):103-18. In general, the section on the role on inflammation is shorter than the other two parts of the triad of dysfunction and could use some more discussing od the role of inflammation in initiating

  1. The new scenario put forward by the authors on line 344-349 still put dysfunctional mature adipocytes at the central stage, and suggest that only after adipocytes have become dysfunctional, they impair adipogenesis by producing peroxidation product – is this theory not also “adipocentric”, putting pre-adipocytes only as a secondary mediator of further dysfunction rather than the initiator? Moreover, if dysfunction is initiated by pre-adipocytes becoming senescent first, as suggested on page 8, this would mean that the peroxidation products have no real role as pre-adipocyte proliferation is already impaired by senescence. The manuscript would benefit from a longer holistic discussion/speculation on how the different theories potentially relate to each other, which would also add to its novelty, by expanding section 6 about future perspectives.

Minor:

  • The citation marks (“…”) are not need when jargon is mentioned, and could be taken away in many places such as on row 200-201 to give one good example
  • Row 208 doubling of Barbara

Senescence in mature adipocytes was recently shown (Li et al, Obesity and hyperinsulinemia drive adipocytes to activate a cell cycle program and senesce Nature medicine 27(11):1941-1953 (2021)), which could be good to add to the references as the current reference for mature adipocytes becoming senescent  (#110, mentioned on row 400-401) is only a review.

Author Response

Reviewer 1

We thank reviewer 1 for hir/her constructive criticisms and important suggestions.

  1. Q. Given the focus of the review and the many mentions of the Adipose tissue Expandability hypothesis, the original reference for it (Virtue & Vidal-Puig: It's Not How Fat You Are, It's What You Do with It That Counts, Plos Biology, 2008) should be references to – I believe it is missing from the current reference list.

  1. In the revised version of the manuscript the reference suggested by the reviewer has been included in the reference list (see p. 5; line 201; ref. n. 42). Moreover, the expandability hypothesis proposed by Vidal-Puig is cited in other three different references (ref. 17; 21 and 41).
  1. Q. The “Expandability hypothesis” is, both in the abstract and on for example on row 121, called novel. However, it has existed for over 10 years in the literature (see reference above) making it not that novel anymore. The current paper would therefore benefit from the authors going through it and reserving the word novel only for concepts that have emerged in the last 1-2 years. One can instead talk about a shift in acceptance or attitude by the field when a theory has been there but not readily accepted in the beginning, if that is what the authors meant?
  1. We thank the reviewer for this criticism. In the revised manuscript we removed the adjective “novel” and rephrased the “expandability hypothesis” accordingly (see p. 4; paragraph 4.1; lines 182-).
  1. Q. According to my interpretation, the original Adipose tissue Expandability hypothesis was, as the authors call it, “adipocentric” and focused on the capacity of adipocytes to store lipids, and less of the role of adipogenesis. Therefore, the concept of defective adipogenesis limiting the adipose tissue’s capacity for expansion is a separate, emerging hypothesis about how obesity and insulin resistance are linked, that has come from the original expandability hypothesis. It seems to me that this concept is different from the original adipose tissue expandability hypothesis, and would therefore require a different name, either already proposed in other papers or laid forward in the present review. The current paper mixes the use of this name for both the original (2008) hypothesis and the renewed, refined one focusing on adipogenesis, making it unclear.
  1. In our review, we hypothesized that insulin-resistance may be a pathological cue initiated in, and sustained by, a dysregulated adipose organ.. In that sense, the concept of “adipocentric” we used was not actually referred to “adipocytes” but to the whole adipose organ. According to the reviewer’s suggestion, in paragraph 4 (see page 4; lines 171-181) we discussed separately the concept of “adipose organ dysfunction” and adipose tissue expandability hypothesis, which has been rephrased and called “revised” (see p. 4, line 182-; paragraph 4.1). Moreover, as also suggested by the Reviewer 2, we divided section 4 into two different paragraphs, which hopefully should better elucidate the concepts of tissue expandability, precursor cell differentiation and adipose dysfunction.
  1. Q. Figure 1 is copied from another publication, as clearly stated in the figure legend. I am however sceptic that this can be done without asking permission, and think a good publication should preferentially contain original artwork, not copied ones.
  1. In the revised manuscript Figure 1 and Figure 2 were completely redrawn. In the previous version of the manuscript, we clearly indicated in the figure legend that Figure 1 was modified (and actually only partly “copied”) from another paper. We were fully aware about the needing of copyright request and permission, as discussed in our correspondence with the Assistant Editor of the Journal.
  1. Q. Row 184-184: doesn’t the SAT do both things (protect against ectopic fat and against spillover to visceral fat)? I suggest changing EITHER on row 184 to BOTH, and OR on row 186 to AND.
  1. In the revised manuscript we changed “either…or” with “both … and” (see p. 5, lines 198-200);
  1. Q. The authors mention that inflammation perturbates adipogenesis – however I think that concept is very much dependent on timing and context, and the authors should include and also discuss the paper from the Sherer lab (Wernstedt Asterholm I, Tao C, Morley TS, Wang QA, Delgado-Lopez F, Wang ZV, et al. Adipocyte inflammation is essential for healthy adipose tissue expansion and remodeling. Cell Metab. 2014;20 (1):103-18. In general, the section on the role on inflammation is shorter than the other two parts of the triad of dysfunction and could use some more discussing of the role of inflammation in initiating
  1. We thank again the reviewer for this important observation. In the revised manuscript we discussed the paper by Scherer’s group (which has been included in the reference list; see ref. 61) and expanded the paragraph about the role of inflammation in AT remodeling (see p. 6; lines 267-283).
  1. Q1. The new scenario put forward by the authors on line 344-349 still put dysfunctional mature adipocytes at the central stage, and suggest that only after adipocytes have become dysfunctional, they impair adipogenesis by producing peroxidation product – is this theory not also “adipocentric”, putting pre-adipocytes only as a secondary mediator of further dysfunction rather than the initiator?

A1. We thank the reviewer for this criticism. In the revised manuscript (section 5) of we focused on the role of oxidative stress as a central player linking adipose plasticity and unhealthy fat expansion with metabolic perturbations. The scenario therein suggested was aimed to explain how the lipid peroxidation products contribute to modify adipose microenvironment. Since these by-products are produced by adipocytes (the main source of lipid storage), the precursor cells became “the victim” of the lipoxidation, as the reviewer argues. On the other hand, as far as lipoxidation is concerned, it is difficult to establish the exact chronology of the cross-talk between precursor cells and mature adipocytes. In other words, we cannot exclude that the formation of oxysterols or hydroxyalchenals also in remote tissue/organs (ie, liver) may occur long before adipocyte became enlarged and dysfunctional. Moreover, data in humans demonstrate that, after one week of high calory diet, oxidative stress in fat actually precedes AT inflammation (Boden G et al. Sci Transl Med 2015; 7:304re7). If this is the case, the impairment of adipogenesis precedes and predicts adipocyte hypertrophy, and the increased secretion of lipid peroxidation products by enlarged fat cells would be a consequence of restricted adipogenesis, which, in turn, sustains a vicious cycle. Nonetheless, since we showed that also at “physiological” levels (1 micromolar) oxysterols may impair APCs viability and adipogenic differentiation, lipoxidation may well induce profound changes in the adiponiche microenvironment by primarily affecting the adipose precursor cells commitment and differentiation towards the adipogenic lineage. In the revised manuscript, in order to better clarify this concept, we rephrased the sentence (see p.8, lines 399-).

Q2. Moreover, if dysfunction is initiated by pre-adipocytes becoming senescent first, as suggested on page 8, this would mean that the peroxidation products have no real role as pre-adipocyte proliferation is already impaired by senescence.

A2. This is another important, but still debated, topic. However, we do not fully agree with the observation of the reviewer that “the peroxidation products have no real role as pre-adipocyte proliferation is already impaired by senescence”. In section 5.2 (p. 9) we discussed about the driving role of lipid peroxidation and its end-products in the induction of a senescent phenotype in different cell types (including stem cells). Thus, the role of increased lipoxidation in fat as a culprit of the higher susceptibility to senescence, especially in adipose precursor cells, appears circumstantiated. Again, as also previously discussed (see point 7; answer A1), we cannot establish the exact chronology between lipoxidation and APCs senescence. However, since data in humans demonstrate that oxidative stress in fat occurs earlier than other adaptive changes (i.e., inflammation) during a high-fat diet (Boden G et al. Sci Transl Med 2015; 7:304re7), the possibility exists that lipoxidation induces senescence in APCs. This opens a novel scenario whereby inflammation, oxidative stress and senescence concur each other in the development of adipose dysfunction also by inducing APCs senescence.

  1. Q. The manuscript would benefit from a longer holistic discussion/speculation on how the different theories potentially relate to each other, which would also add to its novelty, by expanding section 6 about future perspectives.

  1. In the revised manuscript we tried to reconcile the different theories in a holistic discussion/speculation, by expanding section 6 (see pp. 10-11).

 Minor:

  • In the revised manuscript, the citation marks were removed, wherever needed.
  • In row 208 (now at p. 5, line 230 of the revised manuscript) we modified and removed such a mistyping (doubling of Barbara);
  • The reference by Li et al. Nat Med 2021 was added to the reference list (see ref.121, p. 9, line 454).

Reviewer 2 Report

De Fano et al have presented a well written review that covers the consequences of adipose expansion and mechanisms that associate and potentially drive unhealthy fat. The review is well organized and logically steps through aspects and components of physical plasticity, lipid processing and storage, impact of oxidative stress and cellular senescence. While the review is not comprehensive, it does provide support for each aspect addressed. I would suggests some minor considerations to address:

Section 4 is an extensive paragraph that should be separated around lines 204.  The second half more focusing on FAHFAs. It is unclear whether differences in FAHFAs exist between adipose depots and might add some consideration for this. 

Section 5. Might be relevant to address the impact of ROS and/or 4HNE on macrophages. As a role of macrophages in dysfunctional fat seems minimally addressed in this review. 

New figures should be generated. Both figures are minimal adaptations of previously published images and it is unknown whether appropriate approval from the authors or publishers was requested. Additionally, these lack of specific references within the figures fails to highlight aspects of this review (ASC, lipokines, FAHFAs, ROS, 4-HNE, ect...) New figures including these specifically would greatly enhance the readability and make for a better reference/tool for readers.

Author Response

Reviewer 2

We thank reviewer 2 for hir/her constructive criticisms and suggestions.

De Fano et al have presented a well written review that covers the consequences of adipose expansion and mechanisms that associate and potentially drive unhealthy fat. The review is well organized and logically steps through aspects and components of physical plasticity, lipid processing and storage, impact of oxidative stress and cellular senescence. While the review is not comprehensive, it does provide support for each aspect addressed. I would suggest some minor considerations to address:

  1. Section 4 is an extensive paragraph that should be separated around lines 204. The second half more focusing on FAHFAs. It is unclear whether differences in FAHFAs exist between adipose depots and might add some consideration for this. 
  2. A. In the revised manuscript, we separated section 4 into two different paragraphs, according to the suggestion of the reviewer. We did not find data showing differences in the abundance of FAHFAs between different fat depots. Thus, in the revised manuscript we briefly argued about that (see pp 5-6);
  3. Section 5. Might be relevant to address the impact of ROS and/or 4-HNE on macrophages. As a role of macrophages in dysfunctional fat seems minimally addressed in this review.

  1. We thank the reviewer for this important suggestion. In the revised manuscript impact of ROS and lipoperoxidation on macrophages, as well as the role of macrophages in dysfunctional fat was briefly outlined (see p. 8; lines 383-392);

  1. Q. New figures should be generated. Both figures are minimal adaptations of previously published images and it is unknown whether appropriate approval from the authors or publishers was requested. Additionally, these lack of specific references within the figures fails to highlight aspects of this review (ASC, lipokines, FAHFAs, ROS, 4-HNE, ect...) New figures including these specifically would greatly enhance the readability and make for a better reference/tool for readers.

  1. A. In the revised manuscript Figure 1 and Figure 2 were completely modified and redrawn by a cartoonist.

Reviewer 3 Report

This article systematically reviews the characteristics of adipose tissue plasticity and explores the causes and consequences of adipose tissue dysfunction. It is a very meaningful review, and readers will benefit greatly. 

But it is sorry to say that the writing is a little bit obscure and not easy to understand. The structure level of each part of the manuscript is inconsistent, and it is difficult for readers to quickly grasp the key content.

Are there too many uncommon (non-English words) words, such as “accrual”(accumulation), “ deleterious milieu”(environment), “clinical redouts”(??) in the text?

Too many parentheses in the text make the reading not that fluent. Some words, such as hypertrophy and hyperplasia, are explained repeatedly.

Line 24-26: The authors state that “a deleterious milieu either by promoting ectopic lipid overspill in non-adipose targets or by inducing an altered secretion of different adipose-derived hormones” as a “novel paradigm”. Actually, this opinion is not that “novel”. Would it be better to state it in another way?  

Author Response

Reviewer 3

This article systematically reviews the characteristics of adipose tissue plasticity and explores the causes and consequences of adipose tissue dysfunction. It is a very meaningful review, and readers will benefit greatly. But it is sorry to say that the writing is a little bit obscure and not easy to understand. The structure level of each part of the manuscript is inconsistent, and it is difficult for readers to quickly grasp the key content.

We thank reviewer 3 for his/her criticisms and suggestions. In the revised manuscript, in order to increase the “consistency of the work, we extensively modified different parts of the text. Moreover, at the end of each paragraph, a brief summary of the whole topic discussed was provided. We hope that now, the revised version of our work may be more “consistent”. Finally, as claimed by the reviewer 3 and in accordance with the Journal style, we tried to provide more clear and concise key messages, so that the whole picture may be easy to understand for the readers.

  1. Are there too many uncommon (non-English) words, such as “accrual” (accumulation), “deleterious milieu” (environment), “clinical redouts” (??) in the text?

  1. In the revised manuscript we changed the word “redouts” in “readouts” (see p. 2, line 71). We apologize for the mistake due to a misspelling.

1. The word “accrual” is widely used in current literature and accepted in many peer-reviewed journals (see Pediatr Obes 2020; 15(1): e12570 “Body fat accrual trajectories….”; Int J Obes (Lond). 2016; 40(8):1278-85 “Sex differences in the rate of abdominal adipose accrual…..” as good examples). Moreover, the word “accrual” is reported in the Cambridge Dictionary and in the Merriam-Webster Dictionary (https://www.merriam-webster.com/dictionary) as an English “noun”, synonym of “accumulation”. We have no prior reason to believe that this word is not English.

We agree that the word “milieu” is French. However, it has long been used and accepted in many papers and reviews. We hope the reviewer agree.

  1. Too many parentheses in the text make the reading not that fluent. Some words, such as hypertrophy and hyperplasia, are explained repeatedly.

2. In the revised manuscript many parentheses were removed. Moreover, we tried to remove repetition of concepts such as hypertrophy and hyperplasia, according to the reviewer’s suggestion.

  1. Line 24-26: The authors state that “a deleterious milieu either by promoting ectopic lipid overspill in non-adipose targets or by inducing an altered secretion of different adipose-derived hormones” as a “novel paradigm”. Actually, this opinion is not that “novel”. Would it be better to state it in another way?  

3. We thank the reviewer for this suggestion. In the revised manuscript we removed the adjective “novel” and rephrased the expandability hypothesis, as also outlined by the Reviewer 1.